# Fluorescent Polyelectrolyte System to Track Anthocyanins Delivery inside Melanoma Cells

**DOI:** 10.3390/nano11030782

**Published:** 2021-03-19

**Authors:** Raluca Ghiman, Madalina Nistor, Monica Focșan, Adela Pintea, Simion Aștilean, Dumitrita Rugina

**Affiliations:** 1Nanobiophotonics and Laser Microspectroscopy Center, Interdisciplinary Research Institute in Bio-Nano-Sciences, Babes-Bolyai University, 42 Treboniu Laurean, 400271 Cluj-Napoca, Romania; ralughiman@yahoo.com (R.G.); simion.astilean@phys.ubbcluj.ro (S.A.); 2Biochemistry Department, Faculty of Veterinary Medicine, University of Agricultural Science and Veterinary Medicine, 3-5 Calea Manastur, 400372 Cluj-Napoca, Romania; nistor.madalina@usamvcluj.ro (M.N.); apintea@usamvcluj.ro (A.P.); 3Biomolecular Physics Department, Faculty of Physics, Babes-Bolyai University, 1 M. Kogalniceanu, 400084 Cluj-Napoca, Romania

**Keywords:** anthocyanins, polyelectrolytes, microcapsules, delivery, melanoma cells

## Abstract

Over the past decades, there has been a growing interest in using natural molecules with therapeutic potential for biomedical applications. In this context, our aim is focused on anthocyanins (AN) as molecules with anticancer properties that could be used in melanoma local therapies. Due to their susceptibility to environmental changes, current study is based on the design and development of a fluorescent system for carrying and trafficking AN inside melanoma cells. The architectural structure of the proposed system CaCO_3_(PAH)@RBITC@AN reflects a spherical shape, 1080 nm diameter and a solid groundwork CaCO_3_(PAH), on which rhodamine B isothiocyanate (RBITC) fluorophore was firstly added; then, poly(acrylic acid) (PAA) polyelectrolytes and poly(allylamine hydrochloride) (PAH) were successfully deposited. Purified AN from chokeberries were entrapped between PAA layers (rate of 94.6%). In vitro tests confirmed that CaCO_3_(PAH)@RBITC@AN does not affect the proliferation of melanoma B16-F10 cells and proved that their internalization and trafficking can be followed after 24 h of treatment. Data presented here could contribute not only to the existing knowledge about the encapsulation technology of AN but also might bring relevant information for a novel formula to deliver therapeutic molecules or other bio-imaging agents directly into melanoma cells, a strategy that could positively improve tumor therapies.

## 1. Introduction

Nowadays, a lot of effort is invested in the development of innovative delivery systems which can carry molecules with therapeutic potential or with a specific function directly to the tumor site. Anthocyanins (AN) should be taken into consideration as proper therapeutic molecules for delivery systems in medical applications, due to their remarkable ability to reduce tumor cell proliferation and inhibit tumorigenesis [1,2,3]. The presence of a physico-chemical barrier could protect the anthocyanins and diminished their susceptibility to degradation by environmental factors, thus increasing the probability to deliver them undegraded directly into the tumor cells [4,5,6]. In this context, a closed system will protect AN against to an unfavorable environment and will certainly increase their stability, being an efficient strategy for delivering and trafficking them inside melanoma cells [7]. Up to now, AN were successfully entrapped in various systems such as carboxymethyl starch/xanthan gum microcapsules [8], alginate microspheres [9], liposomes [10], hydrogels [11], films [12] and polyelectrolyte microcapsules [5]. 

Polyelectrolyte microcapsules have been intensely investigated in the past years, for their great potential in medicine as drug delivery systems for diagnosis or even for medical imaging. They are considered a new versatile class of (nano)materials, due to a facile way to control their size, shape and permeability, by selecting specific polyelectrolytes as walls and previously establishing the proper number of the deposited layers needed [13]. The most common pair of charged polyelectrolytes used in research is the cationic poly(allylamine hydrochloride) (PAH) and the anionic poly(acrylic acid) (PAA). Addition of polyelectrolytes onto the CaCO_3_ template is usually done by alternating the opposite charged polymers, technique called Layer-by-Layer (LbL) self-assembly [5]. Till now, in few studies AN were successfully loaded in the CaCO_3_ scaffold covered by polymers like chondroitin sulfate and chitosan [5,14].

The novelty of this research article refers to the architecture of a polyelectrolytic system able to entrap anthocyanins and to pass the membrane of melanoma cells. Carrying anthocyanins inside the melanoma cells via microcapsules assures them a high protection toward the degradation environmental factors, being able to fulfill locally their anti-tumoral role. The anthocyanin’s entrapment efficiency of 94.6% was determined based on the high-performance liquid chromatography (HPLC) technique. RBITC fluorophore was grafted between PAH and PAA polymers, in order to follow the internalization and localization of as-formed CaCO_3_(PAH)@RBITC@AN inside melanoma cells by confocal Fluorescence Lifetime Imaging Microscopy (FLIM). The current polyelectrolytic system could be used in medical applications as carrying vehicles for therapeutic molecules like anthocyanins, but also could be a tracking agent for in vitro fluorescence imaging of melanoma.

## 2. Materials and Methods

### 2.1. Materials and Reagents

Reagents used for the microcapsules synthesis: Poly(acrylic acid) (PAA, Mw = 1800 Da), Poly(allylamine hydrochloride) (PAH, Mw = 50.000 Da), Rhodamine B isothiocyanate (RBITC, Mw = 536.08 g/mol), were purchased from Sigma-Aldrich (San Louis, MO, USA). Other reagents such as CaCl_2_ and Na_2_CO_3_ were purchased from Chempur (Jana Lortza, Poland).

Materials used for cell culture studies: Dulbecco’s modified Eagle’s medium (DMEM), fetal bovine serum (FBS), L-glutamine and antibiotics (penicillin and streptomycin mix) were supplied by Gibco (Carlsbad, CA, USA). The cell proliferation reagent WST-1 kit was purchased from Roche Molecular Biochemicals (Indianapolis, IN, USA).

Reagents used for HPLC analysis: cyanidin-3-galactoside (HPLC 90% purity), cyanidin-3-glucoside (HPLC 95% purity), cyanidin (HPLC 95% purity) and LiChrosolv^®^ solvents: acetonitrile, formic acid, methanol were all obtained from Sigma-Aldrich (Darmstadt, Germany). Cyanidin-3-xyloside and cyanidin-3-arabinoside standards were bought from Polyphenols Laboratories (Sandnes, Norway).

### 2.2. Anthocyanins Extraction

AN were extracted from 5 g of chokeberries that were homogenized with acidified methanol (0.01% HCl) by the Ultraturax homogenizer (Ultra-Turrax, Model Miccra D-9 KT, Digitronic GmbH, Bergheim, Germany). The resulted mixture was centrifuged (5000 rpm, 10 min) and the red-colored supernatant was collected. This step was repeated till the sediment became colorless. A purified AN fraction was obtained after the extract loading through a Sep-Pak C18 cartridge. Sugars, acids, and other water-soluble compounds were washed with acidulated water (0.01% HCl). Afterwards, the other polyphenols (less polar than AN) were removed with ethyl acetate. AN were eluted and collected in acidified methanol (0.01% HCl). The resulted purified AN fraction was then evaporated at 35 °C under reduced pressure (Rotavapor R-124, Buchi, Switzerland), being redissolved in acidified water prior to HPLC quantification (total AN quantified as cyanidin-3-galactoside as 1239.3 μg/mL) and was used in the fabrication process of CaCO_3_(PAH)@RBITC@AN. 

### 2.3. Polyelectrolyte System Fabrication 

The fabrication protocol of CaCO_3_(PAH)@RBITC@AN starts with the obtaining of CaCO_3_(PAH) template by a rapid mixing of PAH (1 mg/mL) with CaCl_2_ (50 mM) and Na_2_CO_3_ (50 mM) solutions in equal volumes, at room temperature, on a magnetic stirrer for 30 s. The next step involves the sinking of templates into NaHCO_3_/Na_2_CO_3_ buffer (pH 9.4). Then, the addition of RBITC (1 mg/mL) to the templates’ surface is done under continuously stirring for 4 h, at room temperature. Next, the RBITC excess is removed after a centrifugation (6000 rpm, 5 min) and a few washing steps with NaHCO_3_/Na_2_CO_3_ buffer. Subsequently, the obtained CaCO_3_(PAH) template, now labeled with RBITC were coated with PAA. The PAA polymer (1 mg/mL) was added to the template, and a sonication pulse was added to the solution (amplitude 30%, 1 s), in order to prevent the polyelectrolyte aggregation. Then, the mixture was left to be well-bounded under a gentle stirring for 15 min. The excess of PAA polyelectrolyte was removed by centrifugation (6000 rpm, 5 min), followed by a few washing steps with NaHCO_3_/Na_2_CO_3_ buffer. Cationic AN extracted from chokeberries were linked to PAA layer, in the dark and under agitation for 1h. Last layers added were the opposite polymers PAH and PAA, and the final polyelectrolyte microcapsules CaCO_3_(PAH)@RBITC@AN were obtained. All microcapsules were stored in distilled water at 4 °C protected from light. 

### 2.4. Polyelectrolyte System Characterization 

The free RBITC found in solution was firstly characterized by measuring its UV−Vis absorbance spectrum at room temperature using a Jasco V-670 UV−Vis−NIR spectrophotometer (Tokyo, Japan) with a bandwidth of 2 nm and 1 nm spectral resolution. Fluorescence spectra of as-prepared CaCO_3_(PAH)@RBITC@AN was compared to free RBITC in solution, being collected using a Jasco LP-6500 spectrofluorometer, under the 540 nm excitation, specifically bandwidths of 3 nm in excitation and 3 nm in emission and excitation wavelength. The particle size distribution and zeta-potential of the as-prepared CaCO_3_(PAH)@RBITC@AN were measured using a Nano ZS90 Zetasiser analyzer (Malvern Instruments, Worcestershire, UK) equipped with a He-Ne laser (633 nm, 5 mW). Analyses were performed in triplicate at the scattering angle of 90° and the temperature of 25 °C, reporting their mean value.

The morphology and the mean size of the as-formed CaCO_3_(PAH)@RBITC@AN were examined by TEM imaging, employing a FEI Tecnai F20 microscope. The samples for the TEM studies were prepared by placing a drop of the diluted CaCO_3_(PAH)@RBITC@AN microcapsules solution on the carbon-coated TEM grids.

### 2.5. HPLC-DAD Analysis

Before entrapping AN between the walls of the polyelectrolytes’ microcapsule all were identified and quantified by using high-performance liquid chromatography (HPLC) analysis, on a Shimadzu system equipped with a binary pump delivery system LC-20 AT (Prominence), a degasser DGU-20 A3 (Prominence), diode-array SPD-M20 A UV–VIS detector (DAD) and a Luna Phenomenex C-18 column (5 µm, 25 cm × 4.6 mm). The mobile phase consisted of solvent A—formic acid (4.5%) in bidistilled water and solvent B—acetonitrile. The gradient elution system was: 10% B, 0–9 min; 12% B, 10–17 min; 25% B 18–30 min; 10% B, 31–35 min. All analyses were performed with the flow rate 0.8 mL/min at 35 °C temperature and 520 nm wavelength. The quantitative analysis of AN was based on external calibration using cyanidin-3-glucoside (R^2^ = 0.9900) and standard solution, in the range 10–400 μg/mL.

The entrapment efficiency (EE) was calculated as a difference between total AN used for the entrapment and the non-linked ones. Each sample that could contain AN in the synthesis was collected and analyzed, being nominated as: ANI (supernatant collected after 1h incubation with AN), ANII (sample collected after the first washing step of AN-loaded microcapsules), ANIII (sample collected after the incubation of AN-loaded microcapsules with PAH).

### 2.6. Cell Culture 

The melanoma B16-F10 cell line purchased from American Type Culture Collection (ATCC) was cultured in Dulbecco’s modified Eagle’s medium (DMEM), supplemented with 10% fetal bovine serum (FBS), 1% L-glutamine and 1% antibiotics (penicillin and streptomycin) at 37 °C in a humidified atmosphere containing 5% CO_2_. At the confluence, B16-F10 cells were thawed and resuspended in complete culture medium. 

### 2.7. Proliferation Assay

In the proliferation studies, 8 × 10^3^ B16-F10 cells/well seeded in a 96 well plate, were used to be treated with different concentrations of CaCO_3_(PAH)@RBITC@AN, for 24 h. Then, the WST-1 reagent (15 μL/well) was added for 30 min at 37 °C in 5% CO_2_ atmosphere to reduce the reagent into the colored dye of which absorbance is directly proportional to the number of viable cells. The untreated cells incubated with WST-1 and completed DMEM medium were considered as control (100% cell proliferation). The absorbances were measured at 420 nm with a HT BioTek Synergy microplate reader (BioTek Instruments, Winooski, VT, USA). 

### 2.8. Lifetime Fluorescence Imaging Microscopy 

The internalization process of CaCO_3_(PAH)@RBITC@AN into melanoma B16-F10 cells was monitored by Lifetime Fluorescence Imaging Microscopy (FLIM) technique. Melanoma B16-F10 cells (8 × 10^4^ cells/well, 2-well LabTek Chambered) were treated with 10 microcapsules/cell for 24 h, at 37 °C in 5% CO_2_. After the incubation period, melanoma B16-F10 cells were washed three times with PBS to remove all uninternalized CaCO_3_(PAH)@RBITC@ANs and the cell’s fixation on the substrate by using paraformaldehyde (4%, 20 min, room temperature) was performed. The images were collected with a PicoQuant MicroTime 200 time-resolved inverted confocal fluorescence microscope (IX 71, Olympus) equipped with a Plan N 40x/NA = 0.65 objective. The excitation beam was provided via a fiber coupled, picosecond diode laser head (LDH-D-C 510, 0.55 μW, PicoQuant) operating at 510 nm (20 MHz). The signal collected through the objective was spatially and spectrally filtered by a 50 µm pinhole, long-pass emission filter (HQ519LP, Chroma, Brattleboto, VT, USA) and a photon counting detector module (PDM series, Microphoton devices), connected to a time-correlated single photon counting (TCSPC) electronics (PicoHarp 300, PicoQuant). All images and time-fluorescence decay curves were acquired and analyzed using the SymPhoTime software (version 1.6) provided by PicoQuat. The same configuration was used for the acquisition of FLIM images of control melanoma B16-F10 cells cultured on a 2-well chamber slide. For the acquisition of FLIM images was employed a piezo x−y-scanning table and a PiFoc z-piezo actuator for microscope objective.

## 3. Results

### 3.1. Fabrication and Characterization of Polyelectrolyte Microcapsules

An overview of the entire synthesis process of polyelectrolyte microcapsules is presented in Figure 1A. As can be seen, on a spherical CaCO_3_(PAH) template, by a simple amide coupling reaction, RBITC dye can be easily attached to the positively charge PAH polyelectrolyte of the template. The next layer is represented by the negatively charged PAA, followed by the cationic layer represented by AN. Subsequently, another layer of PAH and PAA were added to obtain the final microcapsule with the following composition CaCO_3_(PAH)-RBITC-(PAA/AN/PAH/PAA), that was further noted as CaCO_3_(PAH)@RBITC@AN. 

The morphological characteristics of CaCO_3_(PAH)@RBITC@AN can be clearly observed in TEM image, which confirm that they are homogeneous, spherical in shape with a rough surface (Figure 1B). Figure 1C brings additional evidence about the free RBITC fluorescence measurement (emission peak at 592 nm, red spectrum) and UV-Vis measurement (absorbance peak at 555 nm (black spectrum)). The overlapping fluorescence emission spectrum of CaCO_3_(PAH)@RBITC@AN (Figure 1C, blue spectrum) to free RBITC fluorescence, demonstrates that RBITC was successfully deposited onto the CaCO_3_(PAH) template, by the shifted emission peak observed at 601 nm, demonstrating thus an efficient entrapment of the fluorescent molecules.

To confirm the addition of the layers RBITC, PAA, AN, PAH, PAA on CaCO_3_(PAH) template with 1050.1 ± 1.5 nm hydrodynamic diameter (calculated for *n* = 3 measurement (Figure 1D, red spectrum), the DLS measurement was done for the final CaCO_3_(PAH)@RBITC@AN microcapsules, resulting an increase in diameter with 30 nm, till 1079.9 ± 2.3 nm (Figure 1D, blue spectrum). To note that a CaCO_3_(PAH) template was also used by Song and coworkers to construct a triple-labeled fluorescent pH-sensitive microcapsule with a diameter of 3200 ± 500 nm, which was successfully monitored inside murine macrophage RAW 246.7 cells [15]. In our recent study polyelectrolyte microcapsules of about 3200 nm hydrodynamic diameter were constructed on CaCO_3_ templates, and each D407 retina cells used in experiments proved to be able to engulf about 3–4 microcapsules [16]

Furthermore, the entire coating procedure to obtain the final CaCO_3_(PAH)@AN(RBITC) microcapsules was subsequently confirmed by zeta potential measurements, by recording the data after each adsorption step (see also the schematic illustration in Figure 1A presenting the coating procedure). As shown in Figure 1E, the zeta potential of the RBITC-labeled CaCO_3_(PAH) presents a positive value at +18, 4 mV (Figure 1E—step 1). Then, the adsorption of anionic PAA layer on CaCO_3_(PAH)@ RBITC shifts the charge toward negative value (−25.3 mV, Figure 1E—step 2). However, when the cationic AN molecules were decorated on the CaCO_3_(PAH)@RBITC-PAA surface, the charge shifts to a positive value (+15,1 mV, Figure 1E—step 3), which indicates that the AN were strongly attached. Subsequently, these as-formed microcapsules were coated again will another layer of cationic PAH (+27 mV, Figure 1E—step 4) The PAH polymeric layer grafted on the cationic AN induced on the particle surface more positive charges, making the microcapsule more stable, as the zeta-potential measurement attest (from +15.1 mV to 27 mV). Of this happening, the cationic PAH polymer molecules bound on the remained vacant sites after the grafting the cationic AN onto the anionic PAA layer. Last layer added onto microcapsules was the anionic PAA (−28 ± 3.2 mV, Figure 1E—step 5), obtaining the final CaCO_3_(PAH)@RBITC@AN fluorescent system carrier for anthocyanins. A negative zeta potential for CaCO_3_ coated-microcapsules with PAA layer has been previously reported in other studies [17]. 

The as-obtained CaCO_3_(PAH)@RBITC@AN microcapsules were kept at 4 °C for one-month and proved to be stable, because all DLS and zeta potential analyses proved no change of their charged surface or hydrodynamic diameter. 

### 3.2. Anthocyanins Entrapment Efficiency 

To know the precise concentration of AN that could be entrapped between the walls of CaCO_3_(PAH)@RBITC@AN, the purified chokeberry extract was analyzed by HPLC (Figure 2A, AN chromatogram). The following AN were identified in the purified extract: cyanidin-3-galactoside (peak 1), cyanidin-3-glucoside (peak 2), cyanidin-3-arabinoside (peak 3), cyanidin-3-xyloside (peak 4) and cyanidin (peak 5) (Figure 2A,B). 

To find out the concentration of AN entrapped, a difference between the initial added concentration (1239 μg/mL expressed as major compound of the extract cyanidin-3-galactoside) (Figure 2, AN sample) and the remained non-entrapped AN (Figure 2, AN I, II, III samples) was calculated. Therefore, all potential non-entrapped AN were collected from three synthesis steps: (i) after 1h incubation of microcapsules with AN solution (denoted further as ANI), (ii) after the first washing step of microcapsules that were incubated with AN (denoted further as ANII), and (iii) after the incubation of microcapsules with PAH polyelectrolyte (denoted further as ANIII). The quantity of AN found in the washing steps of the synthesis represents only 5.4% of total AN added to be entrapped between the walls of CaCO_3_(PAH)@RBITC@AN microcapsules (Figure 2A, ANI, ANII, ANIII). Moreover, it is importantly to emphasize that after adding the other polyelectrolytes layers no other quantifiable losses of AN were seen in the further steps of the synthesis (Figure 2A, ANIII), according to HPLC analysis. This states that AN are tightly kept between the walls of CaCO_3_(PAH)@RBITC@AN once they are entrapped (Table 1). Moreover, this statement can be sustained by the calculated entrapment efficiency of 94.6%. There are other studies in the literature that reported entrapment efficiencies for AN into polysaccharide-based polyelectrolyte microcapsules of 82% [5], or into β-glucan and β-cyclodextrin microcapsules of 45% and 63.25% [8,18].

### 3.3. Cell Proliferation 

Doses of CaCO_3_(PAH)@RBITC@AN from 0 to 40 microcapsules/cell were used to evaluate their effect on B16-F10 melanoma cells’ proliferation. The number of microcapsules for treatment was established by using the cell counting Neubauer chamber under the objective Zeiss Plan-Neofluar 63×/0.75 of Axio Observer A1 Zeiss microscope. 8 × 10^4^ cells/well were exposed to increasing doses of microcapsules for 24 h, prior to WST-1 cell proliferation reagent being added. As can be observed the B16-F10 cells’ proliferation was dose-dependently decreased, but no toxic exerted effect was observed even when the treatment about 40 microcapsules/cell was given. The percentages of the remaining proliferating cells were 96, 93, 92, 86, 86% for doses of 5, 10, 20, 30, 40 microcapsules/cell administered (Figure 3), respectively. 

A similar tendency of decreasing proliferation with no more than 30% (performed with WST-1 reagent) was previously observed in a study done on mouse lymphoma RMA and human lymphoma T1 cells treated with fluorescent antibody-functionalized (PAH/PAA)2 microcapsules [19].

Regarding the anthocyanin’s effect on B16-10 melanoma cells, it was previously published in one of ours articles, where the calculated inhibitory concentration—IC50 of chokeberry-provided anthocyanins was of 352 µg/mL. In the same study is was shown that doses of anthocyanins higher than 300 µg/mL could be cytotoxic for B16-F10 melanoma cells, according to lactate dehydrogenase released assay [20].

### 3.4. FLIM Imaging of Microcapsules 

Unlike intensity-based fluorescence measurements, FLIM is well-suited for detecting local changes in the molecular environment of dyes of interest, even if they have similar emission spectra in the same spectral region. Therefore, FLIM has been extensively employed in the literature to monitor a lot of important intracellular biological processes. Based on its unique advantages, FLIM assay was subsequently employed to map the individual CaCO_3_(PAH)@RBITC@AN microcapsules deposited on microscope substrate, under pulse excitation at 520 nm. The representative confocal images are presented in Figure 4A,B, along with the corresponding lifetime histogram of the FLIM image (Figure 4C). In particular, the collected confocal FLIM images prove that individual CaCO_3_(PAH)@RBITC@AN microcapsules display a strong fluorescence emission with short fluorescence lifetime, as the color scale bar indicates, while the lifetime histogram profile extracted from Figure 4B confirms the fluorescence lifetime of 0.52 ns (Figure 4C). For comparison, the fluorescence lifetime value of free RBITC in water was also evaluated and the fluorescence decay curve was fitted with mono-exponential function fluorescence lifetime of 1.74 ns, which is consistent with the lifetime value reported in the literature by Boens et al. (2007) [21]. Unlike the free RBITC, a significant decrease of the fluorescence lifetime was recorded after its entrapment between polyelectrolytes layers of microcapsules, as stated above, which could be related to an increase of the nonradiative energy transfer to the PAH layer, as a consequence of intermolecular interactions occurring within the complex. The extracted fluorescence lifetime decay is presented in Figure 4D together with the fit function employed (red line). To summarize, considering that our engineered microcapsules present different fluorescence lifetime value compared to endogenous fluorophores intracellularly located—as a source of cellular autofluorescence, generally having fluorescence lifetimes between 2.3 and 4.6 ns [22]—makes them further suitable as biocompatible therapeutic carriers, which can be easily localized inside living cells. 

### 3.5. FLIM Imaging of Internalized Microcapsules in Melanoma Cells

The confocal FLIM, a powerful noninvasive and attractive in vitro imaging technique able to generate contrast images based upon the lifetime of the employed RBITC dye, was subsequently used to monitor the uptake and intracellular localization of our as-engineered CaCO_3_(PAH)@RBITC@AN microcapsules. In this context, B16-F10 melanoma cells treated with 10 microcapsules/cell for 24 h were assessed with imaged by FLIM, under pulse excitation at 520 nm laser line (Figure 5A). Figure 5B represents FLIM image of the B16-F10 cells untreated, considered as control revealing that B16-F10 melanoma cells have a slight autofluorescence and an average lifetime of 2.5 ± 0.02 ns, due to the presence of endogenous fluorophore, such as flavin [22]. Next, in order to track the internalization of CaCO_3_(PAH)@RBITC@AN into B16-F10 melanoma cells after 24h, confocal imaging was performed (Figure 5E). The internalization of CaCO_3_(PAH)@RBITC@AN into B16-F10 melanoma cells recorded two-component lifetime behavior with a low and a high lifetime components of 0.5 ns, characteristic to microcapsules, and 2.66 ns, characteristic to autofluorescence of cells, that can be seen in the lifetime histogram presented in Figure 5F. 

As can be seen in Figure 5E, CaCO_3_(PAH)@RBITC@AN microcapsules undergo the uptake process by B16-F10 melanoma cells after 24 h of treatment and are able to carry AN in the vicinity of nucleus (Figure 5E). Here, the presence of anthocyanins near the nucleus could allow them to kill B16-F10 melanoma cells. It is possible to be activated the NF-κB pathway [2,23]. It is known that the internalization mechanism of micron molecules into living cells is endocytosis [24]. As this mechanism of endocytosis involves the uptake of particles larger than 1000 nm the internalization via clathrin- or caveolae-dependent endocytosis would not be possible. Since B16-F10 cells are non-phagocytic cells and the internalization process was attributed to a non-specific interaction between the particle and the cell membrane, we suggest that macropinocytosis could be the mechanism for internalization of CaCO_3_(PAH)@RBITC@AN. The macropinocytosis process involves the formation of large vesicles that traps a massive “gulp” of extracellular fluid with various particles and takes them into the endocytic vesicles [25]. The same internalization process is proposed in the literature for PSS/PAH polyelectrolyte microcapsules with 5 µm diameter in MDA-MB-435 breast cancer cells, able to engulf around 6 or 7 microcapsules [24]. 

## 4. Conclusions

The findings of this study showed that anthocyanins can be entrapped and efficiently retained between the walls of a polymeric microcapsule, which proved to be stable after a month, at 4 °C storage. Encoded with rhodamine B isothiocyanate dye CaCO_3_(PAH)@RBITC@AN microcapsules are transformed in bio-imaging agents, which allows them to be easily tracked inside B16-F10 melanoma cells. The fluorescent CaCO_3_(PAH)@RBITC@AN are able to carry anthocyanins inside melanoma cells, in the vicinity of the nucleus. The currently developed CaCO_3_(PAH)@RBITC@AN microcapsules are versatile delivery systems for therapeutic molecules, and could be easily adapted for other molecules, but could also be used as bio-imaging agents in melanoma therapy.

## Figures and Tables

**Figure 1 nanomaterials-11-00782-f001:**
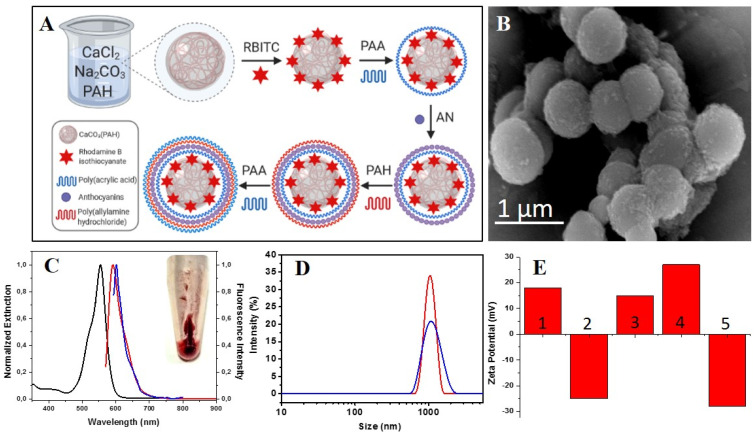
(**A**) Schematic diagram illustrating the fabrication process of CaCO_3_(PAH)@RBITC@AN; (**B**) TEM image of CaCO_3_(PAH)@RBITC@AN microcapsule; (**C**) Spectral overlap between the normalized absorbance (black spectrum) and fluorescence spectra (red spectrum) of free RBITC fluorophore compared to the fluorescence emission of CaCO_3_(PAH)@RBITC@AN (blue spectrum); (**D**) DLS measurements of CaCO_3_(PAH) template (red spectra) and as-obtained CaCO_3_(PAH)@AN(RBITC) microcapsules (blue spectra); (**E**) Zeta potential data recorded after each step: (1) CaCO_3_(PAH)@RBITC, (2) CaCO_3_(PAH)@RBITC-PAA, (3) CaCO_3_(PAH)@RBITC-PAA/AN, (4) CaCO_3_(PAH@RBITC-PAA/AN/PAH, (5) CaCO_3_(PAH@RBITC-PAA/AN/PAH/PAA.

**Figure 2 nanomaterials-11-00782-f002:**
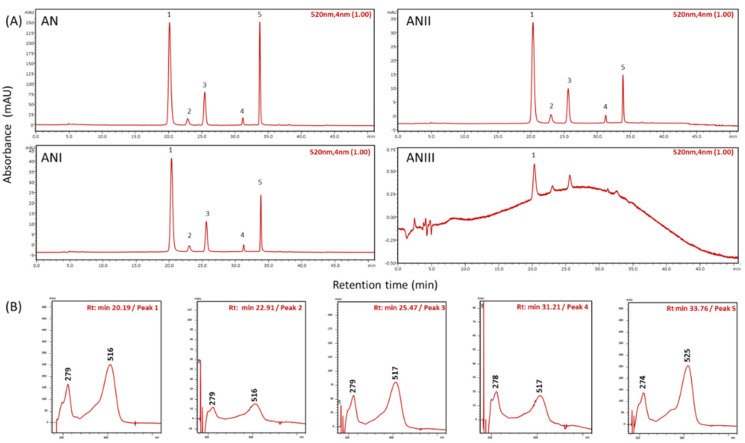
Identification of each anthocyanin from the purified chokeberry extract (AN) and chromatograms recorded in different steps of the CaCO_3_(PAH)@RBITC@AN synthesis (ANI, ANII, ANIII). Identified peaks of chromatograms (**A**): peak1 Cyanidin-3-galactoside; peak2 Cyanidin-3-glucoside; peak3 Cyanidin-3-arabinoside; peak4 Cyanidin-3-xyloside; peak5 Cyanidin and their spectral UV-Vis characteristics (**B**).

**Figure 3 nanomaterials-11-00782-f003:**
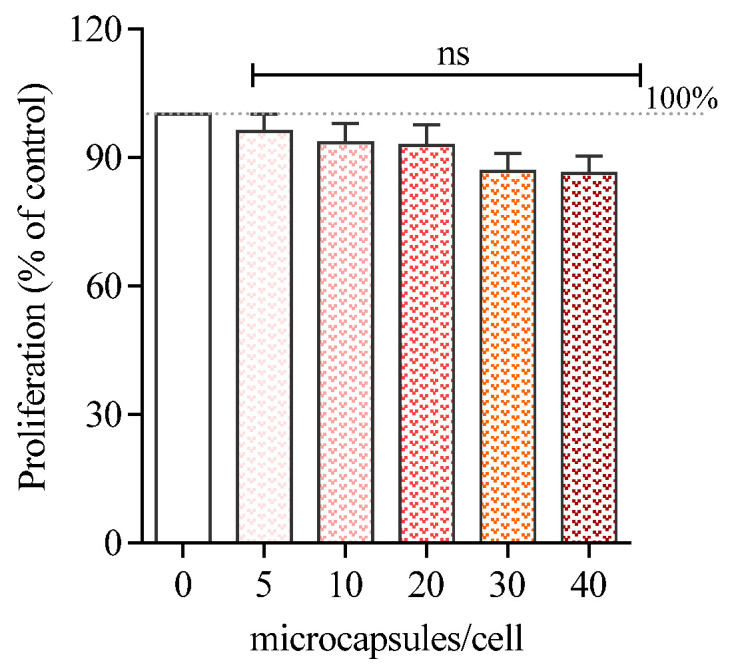
Proliferation of B16-F10 melanoma cells exposed to different concentrations of CaCO_3_(PAH)@RBITC@AN for 24 h. Cell proliferation was assessed by WST-1 assay. Data are expressed as mean ± SEM (*n* = 5). Statistically non-significant differences compared with the untreated control. (one-way Anova; post hoc: Dunnett’s multiple comparison test).

**Figure 4 nanomaterials-11-00782-f004:**
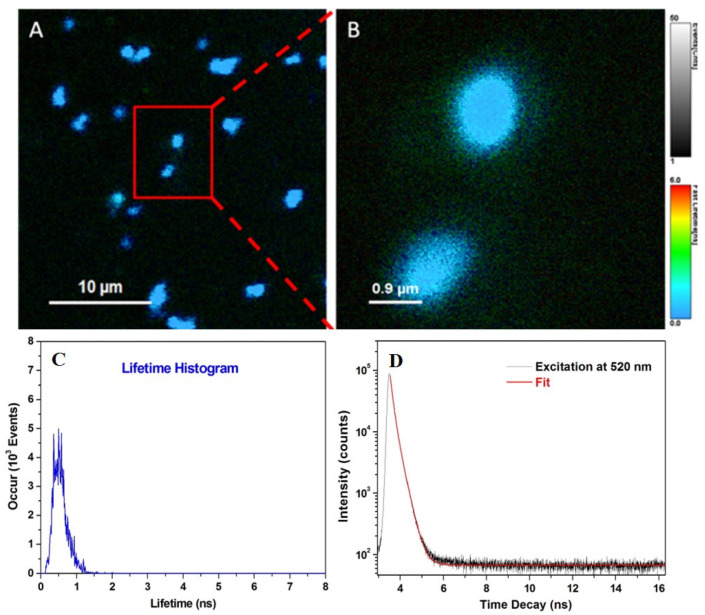
Confocal FLIM images (**A**,**B**) and respective fluorescence lifetime histogram corresponding to FLIM image from B; (**C**) and fluorescence lifetime decay of CaCO_3_(PAH)@RBITC@AN microcapsules dropped on a microscope substrate. Excitation wavelength: 520 nm (**D**).

**Figure 5 nanomaterials-11-00782-f005:**
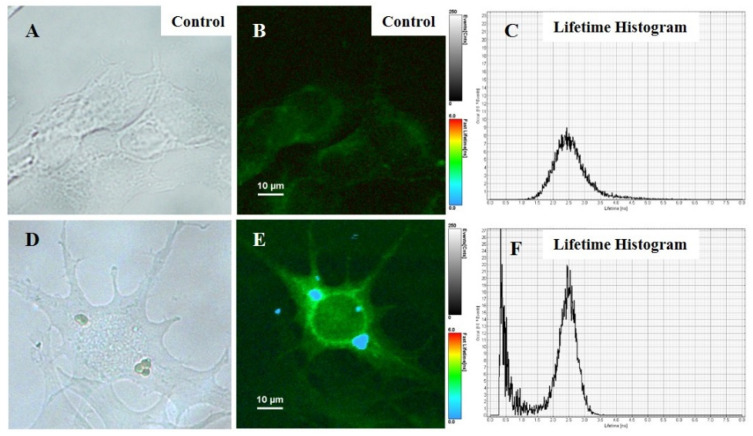
Bright field (**A**,**D**) and FLIM images (**B**,**E**) and their corresponding histograms of CaCO_3_(PAH)@RBITC@AN uptaken (**C**,**F**) by B16-F10 melanoma cells: Untreated (**A**,**B**) and treated with CaCO_3_(PAH)@RBITC@AN (10 microcapsules/cell) (**D**,**E**) for 24 h. The FLIM images denote the fluorescence lifetimes measured at each pixel and displayed as color contrast image. Excitation wavelength: 520 nm.

**Table 1 nanomaterials-11-00782-t001:** Entrapped efficiency (EE) and amounts of entrapped or non-entrapped anthocyanins (AN).

Synthesis Steps	Non-Entrapped AN (µg/mL)	Non-Entrapped AN (%)	EE (%)
ANI	48.02 ± 0.008	3.87 ± 0.006	96.12 ± 0.004
ANII	18.09 ± 0.001	1.46 ± 0.001	98.53 ± 0.009
ANIII	0.77 ± 0.006	0.06 ± 0.002	99.93 ± 0.008
**TOTAL**	**66.89 ± 0.003**	**5.397 ± 0.002**	**94.60 ± 0.002**

The results are expressed as means ± SD.

## Data Availability

The data presented in this study are available on request from the corresponding author.

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
