# Peer review of "Fluorescent Polyelectrolyte System to Track Anthocyanins Delivery inside Melanoma Cells"

_nanomaterials, 2021, doi:10.3390/nano11030782_

Round 1
Reviewer 1 Report
The article deals with the preparation of core-shell microparticles made by a CaCO3(PAH) core coated with various layers of dyes alternated with polyelectrolytes. The authors well describe the synthetic pathway of the microparticles and their internalization in melanoma cells, showing how the complex system is able to reduce cell proliferation by about 30%. In my opinion, the presented architectural structure is very interesting, but it is not clear the role played by loaded anthocyanins. Moreover, while the addition and incorporation of the fluorescent probe (rhodamine B isothiocyanate) were demonstrated by using absorption and emission UV-Vis spectroscopy, the same is absent for anthocyanins. Why? There are only indirect analyses (increment of the hydrodynamic diameter, zeta potential measures, and HPLC analysis of the microparticle washing solution) that suggest the presence of the anthocyanins. Please explain.
What should be the anthocyanins’ role in cell proliferation? This aspect is not very clear. A cell proliferation reduction by 30% was observed, and it was claimed as dose-dependent from the microparticles, but why? Have the authors studied the influence of particles prepared without anthocyanins?
Does antiproliferative action different from cytotoxicity? So, why did the authors said that there isn’t any toxic effect (line 282 par 3.4), but they claim that anthocyanins can kill melanoma cells activating the NF-κB pathway? Please, clarify.
As minor issues:
1) Please add the full names of cited compounds the first time you cited them. For example, add the full name of PAA and PAH at line 50 of the Introduction paragraph.
2) Please improve the English of some sentences in the Introduction paragraph to make them clearer. For example, lines 36-38, 54-46, and 62-64. In particular, regarding the novelty of the study, it should be better to explain better if the novelty regards the high amounts of the anthocyanins onto the microparticles or the system architecture.
3) Please correct the significant digits used for writing hydrodynamic diameters at lines 219-228.
Reviewer 2 Report
The paper” Fluorescent polyelectrolyte system to track anthocyanins delivery inside melanoma cells”, submitted for publication in “Nanomaterials” aims to validate a microcapsule as a nanostructure to carry anthocyanins in melanoma cultured cells, to improve the internalization of these potentially therapeutic molecules. Rhodamine B is also added to the nanostructure as a fluorescing marker to verify and track intracellular internalization of microcapsule. The paper is well written and only some points need to be reconsidered, as a minor revision
Punctual remarks
Page 2, line 57: biocompatible is repeated, and can be removed before “layers”
Page 7, lines 278-279 - The number of microcapsules for treatment is reported to be established by using the cell counting chamber. Given the small dimension of the microcapsules (about 1 um as from Fig. 1B, and Fig. 4 A,B), it should be described the microscope and objective used for observation and counting.
Page 9, the text reports some grammatical and typing errors: line 345 “As we can be seen”; line 346 “melanoma”, line 347 “could allowed”, please correct and also carefully check all along the paper
Page 9, lines 348-349 - the statement: “Surprisingly, Figure 348 5E revealed that one microcapsule was able to enter the melanoma B16-F10 cell’s nucleus” is not convincing. A careful comparative observation of bright field and fluorescence images leads to note that: microcapsules in a perinuclear position can be noticed in the bright field image as darker structures, while the structure suggested to be inside the nucleus is much smaller and bright/transparent, and might only by overimposed to the nucleus or a fragment or a debris similarly to the two small fluorescing spots, one out of the cells, on the left, and one just above the microcapsule at the upper position near the nucleous. Please reconsider these sentences.
Reviewer 3 Report
The paper entitled “Fluorescent polyelectrolyte system to track anthocyanins delivery inside melanoma cells” by Raluca Ghiman, Mǎdǎlina Nistor, Monica Focșan, Adela Pintea, Astilean Simion, Dumitrita Rugina presents a study concerning a novel approach to the encapsulation of anthocyanins into polyelectrolyte microcapsules into polyelectrolyte microcaps. The topic of the work is quite interesting and is in the scope of the Journal, however, some points have to be addressed before publication:
1. Introduction - Sentence that PAH and PAA are the most common used biocompatible polyelectrolytes is confusing, PAH, PAA, and PSS are the most common polyelectrolyte used in research (as model polyelectrolytes) hoverer their biocompatibility is at least debatable. There are other pairs of so-called polyelectrolytes which are commonly used for biomedical application.
2.Part concerning the synthesis and physical-chemical characterization of microcapsules have to be clarified. One of the evidence on the formation of the multilayer structure is the saw-like dependence of the zeta potential of microcapsules measured after each adsorption step. Here in presented results needs that evidence. Your first step was the preparation of CaCO3 (PAH) core, did you measure the zeta potential of your core? It is expected that it should be positively charged, however, results presented in figure 1E indicate that your core is negatively charged. Why? The next step, i.e. labeling with RIBTc should also be evidenced with zeta potential measurements. Depends on the obtained value next polyelectrolyte should be selected. What was the reason to choose PAA as the next layer? And the same story with AN. Adsorption/immobilization of AN should also be evidenced by zeta potential measurements, here also next step depends on the charge of formed microcapsules.
3. In Figure 1D two size distribution curves are presented, what is the difference between them? If the legend for figure 1E is also applicable for figure 1D, why after adsorption of multilayers microcapsules possess a more narrow-sized distribution?
4. You noticed a decrease in fluorescence lifetime for RBITc entrapped in microcapsules shell, is it possible that this decrease is related to bounding RBITc to PAH? Did you measure fluorescently labeled polyelectrolyte in solution?
5. Can you explain why the extremely short fluorescence lifetime of RBITC-labeled polyelectrolyte microcapsules makes them suitable as biocompatible therapeutic carriers? Generally, due to hardware limitations, a shorter lifetime is more difficult to analyze.
Round 2
Reviewer 1 Report
The authors answered to all my issues. Therefore, I think that the paper could be accepted in this new form.
Reviewer 3 Report
There are still some points to discuss.
Reason for choosing PAH (cationic polyelectrolyte) to adsorbed cationic microcapsules CaCO3(PAH)@RBITC-PAH/AN.
What was the driving force of this adsorption?
Since it is an unusual approach, the explanation should be inserted in the manuscript
Round 3
Reviewer 3 Report
Accepted